# Impossibly Good Experts
# And How to Follow Them

**Aaron Walsman[1], Muru Zhang[1], Sanjiban Choudhury[2], Ali Farhadi[1], Dieter Fox[1,3]**
Computer Science and Engineering, University of Washington[1]
Computer Science, Cornell University[2]
NVIDIA[3]
{awalsman,nanami17,ali,fox}@cs.washington.edu
sanjibanc@cornell.edu

## Abstract

We consider the sequential decision making problem of learning from an expert that has access to more information than the learner. For many problems this extra information will enable the expert to achieve greater long term reward than any policy without this privileged information access. We call these experts "Impossibly Good" because no learning algorithm will be able to reproduce their behavior. However, in these settings it is reasonable to attempt to recover the best policy possible given the agent's restricted access to information. We provide a set of necessary criteria on the expert that will allow a learner to recover the optimal policy in the reduced information space from the expert's advice alone. We also provide a new approach called ELF Distillation (**E**xplorer **L**earning from **F**ollower) that can be used in cases where these criteria are not met and environmental rewards must be taken into account. We show that this algorithm performs better than a variety of strong baselines on a challenging suite of Minigrid and Vizdoom environments.

## 1 Introduction

Sequential decision making is one of the most important problems in modern machine learning theory and practice. Reinforcement learning from an environmental reward signal is a powerful but unwieldy tool for attacking these problems. In contrast, imitation learning can be much more sample efficient and empirically easier to train than reinforcement learning, but requires a powerful expert that can either provide an offline dataset of instructions or online supervision. In many practical settings, these experts have access to more information than the learning agent. This can occur when using human demonstrators to train robots that have inferior sensors, or in simulated environments where a synthetic expert uses hidden simulator information to train an agent. In these settings, it is possible that the expert's additional information makes it more powerful than any learning agent that does not have access to the hidden information. We call these experts "Impossibly Good" and show that learning from them using techniques that do not incorporate environmental rewards can cause the agent to drastically under perform the optimal policy in the reduced information space.

For example, consider a simulated robot tasked with retrieving a cell phone in an unknown apartment consisting of multiple rooms. The robot observes the world using a camera and is not given the location of the phone in advance, so it must explore each room in order to find it. Because this is a simulated environment, we can construct an expert that knows not only the location of the phone, but also the exact layout of each room and can compute the shortest path from the robot to the phone. We can then use this expert to construct a large corpus of training data across any number of apartments and phone locations. While these demonstrations may be optimal according to the expert that knows the phone's location, they crucially do not provide any demonstrations of the exploratory behavior that is necessary for the robot which must rely on its more limited sensors. At test time, the robot may need to explore many empty rooms before finding the one that contains the phone, but the expert has always walked directly to the goal and so it has never shown the robot what to do when encountering an empty room. In this case the expert is impossibly good because on average, it can reach the phone much faster than any agent that does not have access to the map, but must explore each room one by one. While we may be able to learn some important skills from this expert, we

are crucially missing demonstrations of other necessary behavior, and so learning from this expert's advice alone may cause the robot to fail.

Our goal in these settings is to find an algorithm that retains the efficiency of imitation learning, while incorporating just enough reward feedback from the environment to achieve success. To address this, we introduce a new technique called ELF Distillation (**E**xplorer **L**earning from **F**ollower). The key insight of this approach is to train one **follower** policy using the advice of the impossibly good expert alone, and then use the estimated long-term value of this policy to drive exploration of a second **explorer** policy using reward shaping. These two policies are trained jointly so that the explorer policy can be used to inform the distribution of states from which the follower must learn.

In order to study these problems, we have constructed a suite of Minigrid(Chevalier-Boisvert et al., 2018) and Vizdoom(Wydmuch et al., 2019) environments that clearly demonstrate the challenges of learning from impossibly good experts. While these are toy problems, they are quite challenging for many strong baselines and related approaches, and allow us to clearly demonstrate the necessary concepts in a setting that avoids confounding implementation details. Code for these experiments can be found at `https://github.com/aaronwalsman/impossibly-good/`.

## 2 RELATED WORK

Recently many authors have identified the problem of learning from experts that have access to more information than the learning agent. This can occur in the self-driving domain, where a human expert may be able to see more than the car's perception system (de Haan et al., 2019; Bansal et al., 2018; Chen et al., 2019). It can also occur robot exploration, where the expert may already know a map of the world (Choudhury et al., 2017; Jain et al., 2021), or in heuristic search where the expert may know which graph edges are valid (Bhardwaj et al., 2017). In some cases it is possible to overcome these issues by considering a history of recent observations instead a single step, but unfortunately this exacerbates the "latching" behavior identified by several practitioners in both self-driving (Muller et al., 2006; Kuefler et al., 2017; Bansal et al., 2018; Codevilla et al., 2019) and natural language processing (Ortega et al., 2021) in which an agent becomes overly fixated on repeating recent actions.

In these partial information settings, Choudhury et al. (2018) showed that interactive imitation learning converges to the QMDP approximation of the expert's policy (Littman et al., 1995). This likely explains the empirical success (Chen et al., 2019; Lee et al., 2020) of such techniques in settings where information naturally reveals itself over time. These interactive imitation learning techniques were originally designed to address covariate shift, a condition in which compounding single-step errors drive the learning agent into states not seen during test time. This effect was first noted by (Pomerleau, 1989) and has long been recognized as one of the most important challenges in imitation learning (Bagnell, 2015). Several approaches have been proposed to address this such as SEARN Daumé et al. (2009), DAgger (Ross et al., 2011) and AggreVaTe (Ross & Bagnell, 2014; Sun et al., 2017). Recent work (Spencer et al., 2021) has shown that covariate shift can be broken into *realizable* settings where off-policy methods such as behavior cloning work well with increasing data, and *non-realizable* settings where on-policy methods with an interactive expert (Ross et al., 2010) or an interactive simulator(Ziebart et al., 2008; Swamy et al., 2021) are necessary.

Some authors (Zhang et al., 2020; Kumor et al., 2021; Ortega et al., 2021) have proposed to address learning in partial information settings using the causal reasoning framework of Pearl et al. (2016), while Swamy et al. (2022) has recently shown that on-policy imitation learning methods can be more effective at recovering the expert's behavior than off-policy approaches in these situations, and has provided conditions under which an agent can asymptotically recover the expert's behavior.

However, in settings where information does not reveal itself, the learner has to actively gather information (Lee et al., 2021). Tennenholtz et al. (2021) examine similar settings but give the learner access to the confounder at test time. Warrington et al. (2021) proposes an asymmetric DAgger algorithm for this setting, but it requires a differentiable model of the expert, which is frequently unavailable. Nguyen et al. (2022) replaces the entropy term in Soft Actor Critic (Haarnoja et al., 2018) with a divergence between the agent and expert policies at each visited state. Weihs et al. (2021) interpolates between the policy gradient and an imitation learning signal using an estimate of how well the agent is able to follow the expert in each state. Our work builds on these ideas by encouraging the agent to visit states where following the expert leads to long term success rather than short-term ability to mimic the expert. This technique uses the tools of pol-

icy distillation (Rusu et al., 2015; Czarnecki et al., 2019) where an agent may learn from a mix of single-step imitation and multi-step reward signals.

Imitation learning from impossibly good experts requires exploratory behavior to find access to hidden information. Deciding how to conduct effective exploration in unknown environments is a problem with a rich history (Kao et al., 1996; Albers & Henzinger, 1997; Yang et al., 2021). Our approach is compatible with a wide range of exploration techniques, but we use the exploration inherent to training stochastic policies with PPO Schulman et al. (2017) in this paper for simplicity.

## 3 IMPOSSIBLY GOOD EXPERTS

In order to study learning from an expert with more information than the learning agent, we use the framework of Partially Observable Markov Decision Processes (POMDPs), (Kaelbling et al., 1998), a generalization of Markov Decision Processes (MDPs) (Sutton & Barto, 2018) to situations where an agent must make decisions using limited observations. The goal in these settings is to sequentially make decisions in discrete time based on feedback from the environment in order to maximize a reward signal. As in an MDP, a POMDP contains an underlying state space $\mathcal{S}$ and action space $\mathcal{A}$. In an MDP, the agent receives the state $s$ from the environment at each time step, but in a POMDP, the learning agent instead receives some possibly lossy and non-Markovian observation $o$ that is a function of the state $o = O(s)$. Because the observations themselves are non-Markovian, the agent must reason over the history of past observations, actions and rewards up until the current time step $\tau_i = \{(o_1, a_1, r_1) \ldots (o_{i-1}, a_{i-1}, r_{i-1}), o_i\}$ referred to as a trajectory.

A learning agent's policy $\pi_{\mathcal{L}}$ is a differentiable function mapping a trajectory to a normalized distribution over actions. We define $\pi_{\mathcal{L}}(\tau)$ as the agent's action distribution after witnessing the history $\tau$. We also define the model class $\Pi_{\mathcal{L}}$ to be the set of all possible learnable policies. During training we assume access to an expert policy $\pi_{\mathcal{E}}$ which is a non-differentiable function that also maps a trajectory to a normalized distribution over actions.

In order to reason about agents and experts that have different information, we consider a POMDP with two separate observation functions. The first $o_{\mathcal{L}} = O_{\mathcal{L}}(s)$ produces an observation that the agent sees, while the second $o_{\mathcal{E}} = O_{\mathcal{E}}(s)$ produces an observation that the expert sees. This allows us to reason about a trajectory in the underlying state space $\tau_{\mathcal{S}}$ and map it to a corresponding trajectory of observations for the agent $\tau_{\mathcal{L}}$ and a separate trajectory of observations for the expert $\tau_{\mathcal{E}}$.

$$\tau_{\mathcal{S}} = \{(s_1, a_1, r_1) \ldots (s_N, a_N, r_N)\}$$
$$\tau_{\mathcal{L}} = \{(O_{\mathcal{L}}(s_1), a_1, r_1) \ldots (O_{\mathcal{L}}(s_N), a_N, r_N)\}$$
$$\tau_{\mathcal{E}} = \{(O_{\mathcal{E}}(s_1), a_1, r_1) \ldots (O_{\mathcal{E}}(s_N), a_N, r_N)\}$$

The observational trajectory $\tau_{\mathcal{L}}$ represents what the learning agent observes while interacting with the environment, while the state trajectory $\tau_{\mathcal{S}}$ represents the ground truth state of the world which is unknown. For non-deterministic observation functions we use $O_{\mathcal{L}}(o_{\mathcal{L}}|s)$ and $O_{\mathcal{L}}(\tau_{\mathcal{L}}|\tau_{\mathcal{S}})$ to represent the probability of a state $s$ or state space trajectory $\tau_{\mathcal{S}}$ producing an observation $o_{\mathcal{L}}$ or observation trajectory $\tau_{\mathcal{L}}$.

We assume a model-free setting in which the states, observation functions, transition dynamics and reward dynamics are all unknown to the learning algorithm. Instead during training, the algorithm interacts with the environment, and receives observations and a scalar reward in the form $\{o_{\mathcal{L}}, o_{\mathcal{E}}, r\}$ with the assumption that $o_{\mathcal{L}}$ and $o_{\mathcal{E}}$ were generated from the same unknown state $s$. The learning algorithm also has access to an expert $\pi_{\mathcal{E}}$ that can produce advice in the form of a suggested action $a_{\mathcal{E}} \sim \pi_{\mathcal{E}}(\tau_{\mathcal{E}})$. At test time the agent will not receive the expert's observations $o_{\mathcal{E}}$ or have access to the expert's advice $a_{\mathcal{E}}$, and so it must learn to make decisions using trajectories of $o_{\mathcal{L}}$ alone.

We consider the episodic setting with a set of terminal states $s^{term}$. When a terminal state is reached, the environment informs the agent that the episode has ended using a special observation $o^{term}$ and resets $s$ using an unknown initial state distribution $S^{init}$. We write $\tau_{\mathcal{L}} \sim \pi_{\mathcal{L}}$ to refer to a trajectory generated by initializing according to $S^{init}$ and repeatedly sampling actions from $\pi_{\mathcal{L}}$ until a terminal state is reached. We also write $\rho_{\pi_{\mathcal{L}}}(o_{\mathcal{L}})$ and $\rho_{\pi_{\mathcal{L}}}(\tau_{\mathcal{L}})$ as the probability of observing $o_{\mathcal{L}}$ and $\tau_{\mathcal{L}}$ respectively when acting according to $\pi_{\mathcal{L}}$ from the initial state distribution. We will also use $P_{\pi_{\mathcal{L}}}$ to refer to the set of trajectories $\tau$ that have nonzero probability when acting according to $\pi_{\mathcal{L}}$ from the initial state distribution.

The return of a trajectory $G(\tau) = \sum_{i=1}^{|\tau|} r_i$ is the sum of all rewards encountered during that trajectory. The value function of a policy $V_{\pi_{\mathcal{L}}}(s) = \mathbb{E}_{\tau \sim \pi_{\mathcal{L}}|s_1=s} G(\tau)$ is the expected value of the return of

a trajectory that starts in state $s$ and continues by sampling actions according to $\pi_{\mathcal{L}}$. We write $V_{\pi_{\mathcal{L}}}^{init}$ to be $\mathbb{E}_{s \sim S^{init}} V_{\pi_{\mathcal{L}}}(s)$, the expected value when acting according to $\pi_{\mathcal{L}}$ after starting from states sampled from the initial state distribution. The goal of the learning algorithm is to learn a policy $\pi_{\mathcal{L}}(\tau_{\mathcal{L}})$ using the reward signal $r$ and expert advice $a_{\mathcal{E}}$ during training in order to maximize $V_{\pi_{\mathcal{L}}}^{init}$ during test time. An agent-optimal policy $\pi_{\mathcal{L}}^*$ is one that achieves the greatest value when starting from the initial state distribution compared to all other policies in the model class $V_{\pi_{\mathcal{L}}^*}^{init} \geq V_{\pi_{\mathcal{L}}}^{init} \, \forall \, \pi_{\mathcal{L}} \in \Pi_{\mathcal{L}}$. Note that while this agent-optimal policy is the best policy achievable with limited information, it may still underperform the expert $\pi_{\mathcal{E}}$ that has more information.

We now have the tools to formally state the impossibly good criterion. We say an expert $\pi_{\mathcal{E}}$ is impossibly good iff

$$V_{\pi_{\mathcal{E}}}^{init} > V_{\pi_{\mathcal{L}}^*}^{init} \tag{1}$$

This means that an expert is impossibly good if in expectation it achieves a greater value from the initial state distribution than the best policy in the model class $\Pi_{\mathcal{L}}$.

## 4 How Not To Follow Them

In this section we consider learning from an impossibly good expert using **Behavior Cloning** and **DAgger**, two common techniques that use expert demonstrations as a learning signal. We first describe these methods, then show examples designed to demonstrate their success and failure modes when learning from impossibly good experts. We then provide formal criteria for determining when these methods are capable of finding the agent-optimal policy $\pi_{\mathcal{L}}^*$ from demonstrations provided by $\pi_{\mathcal{E}}$. Note that Swamy et al. (2022) have also recently provided conditions under which an agent with restricted information may achieve expert level performance from demonstrations. Rather than considering cases where expert performance may be achieved, our goal is instead to give conditions for learning the best policy possible in cases where achieving expert performance is impossible. Finally we show that in some environments, the agent-optimal policy $\pi_{\mathcal{L}}^*$ cannot be recovered from expert advice alone.

**Behavior Cloning** is an off-policy technique in which the expert $\pi_{\mathcal{E}}$ generates a static dataset of trajectories prior to training. The algorithm then iteratively samples batches of state-action transitions from this dataset and trains the learning agent to match the expert's demonstrations. Behavior Cloning is known to suffer from covariate shift (Ross et al., 2011), a condition in which errors made by the model during test time can quickly take the agent outside of the distribution of states encountered during training. Despite this limitation, Behavior Cloning enjoys widespread popularity due to its simplicity and ability to perform well on some problems (Spencer et al., 2021).

**DAgger** Ross et al. (2011) is an on-policy method designed to overcome the covariate shift of Behavior Cloning. Recently Swamy et al. (2022) has shown that on-policy methods such as DAgger can improve an agent's performance when learning from an expert with more information. Instead of generating a static dataset by acting according to the expert $\pi_{\mathcal{E}}$, DAgger iteratively collects batches of data by acting according to the learning agent $\pi_{\mathcal{L}}$ and simultaneously recording the expert's advice in each visited state. After collecting a batch of data, the agent is trained to replicate the expert's advice on all data collected so far. This allows the agent to train on states that may not be visited by the expert, but are necessary to recovering from mistakes made by the learner.

Diagrams of the examples in this section are found in Figure 1. In these environments, the state space, action space and state-action transition function consist of a directed graph of nodes representing the location of the agent. The state space also contains a random variable $X$ that is uniformly sampled from $\{0, 1\}$ at the start of each episode and remains fixed until termination. The expert observes the value of $X$ at the start of the episode, while the agent only observes $X$ after reaching certain nodes in the graph. Transitions between nodes are deterministic, and both expert and agent observe the node they currently occupy. In all examples there are two terminal nodes, one which produces a reward of 1 when $X = 0$ and the other which produces a reward of 1 when $X = 1$. The agent also receives a small reward penalty $-\epsilon$ at each step to encourage faster solutions.

Due to the deterministic transition function and the fact that $X$ is fixed for each episode, we can simplify $\pi_{\mathcal{L}}$ to be a function of the current node and the remembered value of $X \in \{0, 1, ?\}$ rather than an entire trajectory of observations. We also assume the expert $\pi_{\mathcal{E}}$ acts optimally according to its observations.

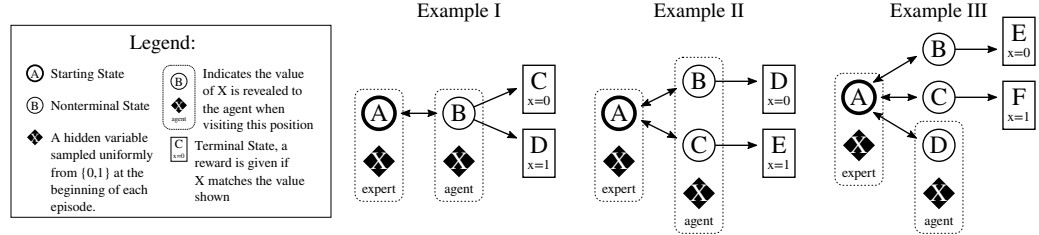

Figure 1: Three example environments. An agent may travel through states by taking actions as indicated by the arrows. The reward when reaching a terminal state depends on a hidden variable X that is revealed to the agent if it visits the indicated locations.

First consider Example I. Here, the agent sees the value of $X$ as soon as it visits $B$. This allows the agent-optimal policy $\pi_{\mathcal{L}}^*$ to always know which terminal reward to transition to from B. An agent in this example must learn distributions for observations $\pi_{\mathcal{L}}(A, ?)$, $\pi_{\mathcal{L}}(B, 0)$ and $\pi_{\mathcal{L}}(B, 1)$. Because the expert $\pi_{\mathcal{E}}$ takes the same path as the agent-optimal policy, it will generate correct training examples for these distributions. Using Behavior Cloning to match these examples will recover $\pi_{\mathcal{L}}^*$.

Example II in Figure 1 shows a case where Behavior Cloning fails. To see this, note that when generating data, the expert will travel from A to B when $X = 0$ and from A to C when $X = 1$. Under the agent's observation function $O_{\mathcal{L}}$, these correspond to $(A, ?)$, $(B, 0)$ and $(C, 1)$. Now consider the agent-optimal behavior in this example. Because of its limited information, the agent must take the first step blindly, and so it will sometimes reach either $(B, 1)$ or $(C, 0)$. In these cases, the optimal behavior is to backtrack to $A$ where it will encounter either $(A, 0)$ or $(A, 1)$, remembering the value of $X$. From these observations, the agent now has enough information to guarantee reaching $(B, 0)$ or $(C, 1)$ and achieve the terminal reward. Note though that data generated by acting according to the expert did not provide demonstrations for what to do when observing $(B, 1)$ or $(C, 0)$ so we cannot hope to recover this optimal behavior.

On the other hand, we can show that DAgger can recover $\pi_{\mathcal{L}}^*$ in this example. Now data is generated by the learning agent $\pi_{\mathcal{L}}$, and as we just saw, it has no way to decide whether to transition to B or C at the first step. This means that unlike Behavior Cloning, the observations $(B, 1)$ and $(C, 0)$ will exist in the training set. For these observations, the optimal behavior according to the expert $\pi_{\mathcal{E}}$ is to backtrack to $A$ and continue down the opposite branch so these labels will be provided along with the corresponding observations. These labels are also optimal according to $\pi_{\mathcal{L}}^*$, so this extra data will allow DAgger to recover the agent-optimal policy.

Finally Example III in Figure 1 shows a case where DAgger will fail to recover $\pi_{\mathcal{L}}^*$. Now the problem is not only data coverage, but also the instructions received by the expert. Similar to Example II, the expert will always tell the learner to transition to $B$ or $C$ from the initial observation $(A, ?)$. However, due to the information asymmetry, the agent-optimal policy $\pi_{\mathcal{L}}^*$ must first visit $D$ in order to discover the value of $X$ before backtracking to $A$ and continuing to $B$ or C as appropriate. Even if some uncertainty in the agent's policy causes it to visit $D$ while generating data, the labels for $(A, ?)$ will never tell the agent that this is the correct behavior.

These examples have given us two intuitive conditions that must be met in order for these methods to recover $\pi_{\mathcal{L}}^*$. The first is that all observations that are encountered when acting according to the agent-optimal policy $\pi_{\mathcal{L}}^*$ must be encountered during training. The second condition is that the expert $\pi_{\mathcal{E}}$ must instruct the agent to take correct actions according to $\pi_{\mathcal{L}}^*$. We now generalize these intuitions into a formal set of conditions that must be met in order for these methods to recover the agent-optimal policy $\pi_{\mathcal{L}}^*$ in a simplified learning model.

Consider a dataset $D$ generated by acting according to an arbitrary generation policy $\pi_D$ and sampling labels $a_{\mathcal{E}}$ from $\pi_{\mathcal{E}}$. Also consider a learning policy $\widetilde{\pi}_{\mathcal{L}}$ that memorizes the empirical distribution of expert recommendations for each trajectory $\tau_{\mathcal{L}}$ in the dataset $D$.

$$\widetilde{\pi}_{\mathcal{L}}(a|\tau_{\mathcal{L}}) = \frac{\sum_i^{|D|} \mathbb{1}_{\tau_{\mathcal{L}i}=\tau_{\mathcal{L}}, a_{\mathcal{E}i}=a}}{\sum_i^{|D|} \mathbb{1}_{\tau_{\mathcal{L}i}=\tau_{\mathcal{L}}}} \tag{2}$$

**Theorem 1.** *The empirical policy $\widetilde{\pi}_{\mathcal{L}}$ will recover $\pi_{\mathcal{L}}^*$ as $|D| \to \infty$ iff the following conditions hold:*

1. *(Coverage)* $\rho_{\pi_D}(\tau_{\mathcal{L}}^*) \neq 0 \; \forall \; \tau_{\mathcal{L}}^* \in P_{\pi_{\mathcal{L}}^*}$

2. *(Correctness)* $\mathbb{E}_{\tau_{\mathcal{S}} \sim \pi_D} \pi_{\mathcal{E}}(O_{\mathcal{E}}(\tau_{\mathcal{S}})) O_{\mathcal{L}}(\tau_{\mathcal{L}}^* | \tau_{\mathcal{S}}) = \pi_{\mathcal{L}}^*(\tau_{\mathcal{L}}^*) \; \forall \; \tau_{\mathcal{L}}^* \in P_{\pi_{\mathcal{L}}^*}$

The proof is in Appendix A.

The first condition states that all policy input trajectories that have nonzero probability of being visited under the agent-optimal policy $\pi_{\mathcal{L}}^*$ must also have non-zero probability under the dataset sampling policy $\pi_D$. This corresponds to the intuition developed from the examples above that during training we must visit all states that the agent-optimal policy needs to learn about.

The second condition states that the distribution of expert advice for the state trajectories $\tau_{\mathcal{S}}$ which map to agent observation trajectories $\tau_{\mathcal{L}}^*$ must be equal to $\pi_{\mathcal{L}}^*(\tau_{\mathcal{L}}^*)$ for all trajectories with nonzero probability of being visited under the agent-optimal policy $\pi_{\mathcal{L}}^*$. This corresponds to the intuition developed earlier that an expert must tell the agent to act according to $\pi_{\mathcal{L}}^*$ in order to recover $\pi_{\mathcal{L}}^*$.

Next we show it may not be possible to learn the agent-optimal policy $\pi_{\mathcal{L}}^*$ from expert demonstrations alone regardless of the dataset generation policy $\pi_D$.

**Theorem 2.** *There exist environments with impossibly good experts that violate the correctness condition in Theorem 1 regardless of the dataset distribution policy $\pi_D$.*

*Proof.* Example III in Figure 1 is a proof by example.

In this environment $\pi_{\mathcal{L}}^*(A \to D | (A, ?)) = 1.0$. However, $\pi_{\mathcal{E}}(A \to D | \tau_{\mathcal{E}}) = 0 \; \forall \; \tau_{\mathcal{E}}$ so there is no $\pi_D$ that can sample states mapping to $\tau_{\mathcal{E}}$ that will cause $\pi_{\mathcal{E}}(\tau_{\mathcal{E}})$ to produce the labels required to learn $\pi_{\mathcal{L}}^*$. ∎

This result shows that in some cases it is impossible to learn the agent-optimal policy $\pi_{\mathcal{L}}^*$ from the labels generated by an impossibly good expert alone, regardless of the rollout policy $\pi_D$ that generates the dataset.

## 5 How to Follow Them

In the previous section, we have shown that it is possible to construct an environment with an impossibly good expert such that the learner cannot recover the agent-optimal policy when learning from the expert's advice, regardless of the dataset generating distribution $\pi_D$. Paradoxically, this means we must treat advice from these experts as sub-optimal from the agent's perspective, even though they come from an expert that can achieve higher reward than any policy we can learn. In light of this result, it is natural to ask if we can improve this situation by incorporating environmental reward into our learning algorithm.

The literature on imitation learning (Rajeswaran et al., 2017) and policy distillation (Rusu et al., 2015) provides many useful tools that can be used to learn from sub-optimal experts. We describe these methods using the unifying distillation framework of Czarnecki et al. (2019). In this setting, distillation algorithms consist of two steps. The first step rolls out a set of $N$ state-action-reward transitions $D = \{(\tau_1, a_1, r_1) \dots (\tau_N, a_N, r_N)\}$ using a dataset generation policy $\pi_D$ ($q_\theta$ in Czarnecki et al. (2019)). In the second step, the parameters $\theta$ of the learner's policy $\pi_{\mathcal{L}}$ are then updated in proportion to

$$\mathbb{E}_{\pi_D} \left[ \sum_{i=1}^{|D|} -\nabla_\theta \log \pi_{\mathcal{L}}(a_i | \tau_i) \widehat{R}_t + \nabla_\theta l(\pi_{\mathcal{L}}(a_i | \tau_i), \pi_{\mathcal{E}}(a_i | \tau_i)) \right] \quad (3)$$

These steps are repeated until performance converges or some other stopping criterion is met. The update rule in Equation 3 can be thought of as mixing a reinforcement learning objective based on a non-differentiable multi-step return $\widehat{R}_t = \sum_{i=t}^{|\tau|} \hat{r}_i$ with a single-step differentiable loss function $l$. The reward term $\hat{r}_i$ can be the environmental reward $r_i$, or any other term we choose that assigns greater value to some behavior that we want to encourage. The loss $l$ is more restrictive because it must be differentiable, but provides a powerful way to inject direct supervision into the training process. By varying our choice of $\pi_D$, $\hat{r}_i$ and $l$ we can arrive at several different distillation algorithms as shown in Table 1. Here the functions $H(\pi_{\mathcal{E}}, \pi_{\mathcal{L}})_i$ and $KL(\pi_{\mathcal{E}}, \pi_{\mathcal{L}})_i$ respectively refer to the cross entropy and KL divergence between the expert and agent for the current observational history $\tau_i$.

Table 1: Algorithms.

| Algorithm | $\pi_D$ | $l$ | $\widehat{r}_i$ |
|---|---|---|---|
| Policy Gradient | $\pi_{\mathcal{L}}$ | 0 | $r_i$ |
| Teacher Distill | $\pi_{\mathcal{E}}$ | $H(\pi_{\mathcal{E}}, \pi_{\mathcal{L}})_i$ | 0 |
| On-Policy Distill | $\pi_{\mathcal{L}}$ | $H(\pi_{\mathcal{E}}, \pi_{\mathcal{L}})_i$ | 0 |
| On-Policy Distill+R | $\pi_{\mathcal{L}}$ | $H(\pi_{\mathcal{E}}, \pi_{\mathcal{L}})_i$ | $r_i$ |
| N-Distill +R | $\pi_{\mathcal{L}}$ | $H(\pi_{\mathcal{E}}, \pi_{\mathcal{L}})_i$ | $-H(\pi_{\mathcal{E}}, \pi_{\mathcal{L}})_{i+1} + r_i$ |
| Expert Matching Reward +R | $\pi_{\mathcal{L}}$ | 0 | $\alpha \mathbb{1}_{a_i = a_{\mathcal{E}i}} - \beta \mathbb{1}_{a_i \neq a_{\mathcal{E}i}} + r_i$ |
| ADVISOR | $\pi_{\mathcal{L}}$ | $H(\pi_{\mathcal{E}}, \pi_{\mathcal{L}})_i w_i$ | $r_i(1 - w_i)$ |
| ELF | $\pi_{\mathcal{L}}$ | 0 | $r_i + v_{\mathcal{F}}(\tau_{i+1}) - v_{\mathcal{F}}(\tau_i)$ |

**Policy Gradient** is a general class of reinforcement learning algorithms (Williams, 1992; Kakade, 2001; Schulman et al., 2015; 2017). Applying policy gradient methods to our problems essentially throws away the expert advice and learns from environmental rewards alone.

**Teacher Distill** and **On-Policy Distill** are the names used by Czarnecki et al. (2019) for versions of the Behavior Cloning and DAgger Ross et al. (2011) algorithms adapted for the distillation framework where data is continually generated online. These algorithms do not include an $\hat{r}_i$ term, and so will not be able to recover $\pi_{\mathcal{L}}^*$ in all cases.

**On-Policy Distill+R** Combines the single step loss term from On-Policy Distill with the environmental reward used in Policy Gradient. The motivation is that if On-Policy Distill attempts to follow the expert, and Policy Gradient seeks high environmental reward, then combining them seeks to do both.

**N-Distill+R** is also from Czarnecki et al. (2019) and is similar to COSIL (Nguyen et al., 2022), but adapted to the distillation framework. Unlike COSIL, N-Distill+R uses cross entropy rather than the KL Divergence, but these are identical for deterministic experts, which is the setting we assume in our experiments. The other difference is that N-Distill+R uses the cross entropy term from the first time step as a differentiable loss rather than rolling it into the reward signal, which improves performance. This method is also similar to On-Policy Distill+R except that it augments the reward using future agreement with the expert. This encourages the model to not only take actions that the expert immediately agrees with, but also to seek out states in which it is easy to agree with the expert in the future.

**Expert matching Reward+R** removes the single-step loss term and instead assigns a fixed scalar $\alpha$ to the reward $r_i$ when the agent's actions agree with the expert and negative reward $-\beta$ when the actions disagree.

**Teacher Distill + PPO** is similar to DAPG (Rajeswaran et al., 2017) and first trains an agent with Teacher Distill for a fixed number of time steps, then refines this policy using reinforcement learning. This is a popular technique used to learn robot behavior from human demonstrations.

All methods mentioned so far that attempt to learn from both the expert advice and the reward signal suffer from a common problem: they all attempt to balance reward seeking and expert following behavior uniformly across all sequences of observations, and do not attempt to explicitly discover where following the expert yields high reward and where it does not.

**ADVISOR**, from Weihs et al. (2021) attempts to address this by dynamically interpolating between imitation and reinforcement learning signals using a weighting factor $w_i = e^{-\alpha KL(\pi_{\mathcal{E}}, \pi_{aux})_i}$, where $\pi_{aux}$ is an auxiliary policy trained to follow the expert in every state using the information available to the agent. This weighting factor estimates how closely the agent can follow the expert given the information it has at the current time step. Unfortunately, the ability to follow the expert at a given state is not strictly indicative of states that require exploration. An example demonstrating how this can lead to suboptimal behavior is shown in Appendix E.

**ELF Distillation** is our new technique which overcomes the limitations of the previous methods. Pseudocode is shown in Algorithm 1. The key insight is to train two policies jointly: a *follower* $\pi_{\mathcal{F}}$ which attempts to learn how to follow the expert, and an *explorer* $\pi_{\mathcal{L}}$ that attempts to maximize environmental reward using the follower's value function as reward shaping.

The follower is trained using a distillation that samples data according to a switching policy that rolls out experience according to the explorer $\pi_{\mathcal{L}}$ for a random number of time steps, then allows the follower $\pi_{\mathcal{F}}$ to take over. This rollout behavior is also used in AggreVaTe Ross & Bagnell (2014) and allows us to learn to follow the expert from states visited by the explorer that would not normally be visited by the follower. We also train a value function $v_{\mathcal{F}}$ on the second half of the trajectory that is rolled out according to the follower. The follower's distillation uses cross entropy to the expert's advice as the single-step loss function $l$ with no reward term.

The explorer is trained using policy gradients with reward shaping that encourages the explorer to visit states with high value according to the follower's learned value function. The reward term $\hat{r}_i = r_i + v_{\mathcal{F}}(\tau_{i+1}) - v_{\mathcal{F}}(\tau_i)$ uses the follower's learned value function $v_{\mathcal{F}}$ as potential-based reward shaping Ng et al. (1999). This is referred to as Teacher V Reward in Czarnecki et al. (2019) and is related to Sun et al. (2018). The single-step loss function is 0 unless the follower's value function $v_{\mathcal{F}}(\tau)$ is greater than a target value $v$ in which case it uses cross-entropy to the follower's distribution. This accelerates learning in states where the follower has already learned to reach high returns.

Training proceeds in an alternating fashion, first training the follower on new data generated by the switching policy, then training the explorer on new data generated using the explorer (on-policy). It is necessary to keep training the follower because the explorer may reach states in the middle of training that would not be visited otherwise, and it is important to build accurate value estimates for them as well.

At test time, the follower is no longer necessary and can be discarded. The only purpose of the follower is to discover where the expert is reliable and where it isn't in order to overcome the limitation of trying to find good global mixing rules for reward seeking and expert following behavior. In this way the follower policy is similar to the auxiliary policy in ADVISOR, but it is used by the algorithm in a much different way. Rather than guiding exploration using the follower's ability to replicate the expert in a given state, we instead use the follower's value estimate, which encourages the learner to visit states where following the expert leads to high long-term reward. This avoids important failure cases such as the one shown in Appendix E.

---

**Algorithm 1** ELF Distillation

---

**Require:** $N_{steps}$, Expert $\pi_{\mathcal{E}}$, Horizon $T$, Dataset Sizes $N_{follow}$, $N_{explore}$, Target value $v$.
   Initialize follower $\pi_{\mathcal{F}}$.
   Initialize follower value estimator $v_{\mathcal{F}}$.
   Initialize explorer $\pi_{\mathcal{L}}$.
   **for** $i = 1$ **to** $N_{steps}$ **do**
      Initialize $D_{follow} = \{\}$.
      **for** $j = 1$ **to** $N_{follow}$ **do**
         Choose a uniform random switching time $t \in \{1, \ldots, T\}$.
         Add new trajectory $\tau$ to $D_{follow}$ by sampling $\tau_{1\ldots t} \sim \pi_{\mathcal{L}}$ and $\tau_{t\ldots T} \sim \pi_{\mathcal{F}}$.
      **end for**
      Train $\pi_{\mathcal{F}}$ to match $\pi_{\mathcal{E}}$ for all transitions in $D_{follow}$ using cross entropy.
      Train $v_{\mathcal{F}}$ to match the returns from all transitions in $D_{follow}$ after the switching time $t$.
      Initialize $D_{explore} = \{\}$.
      **for** $j = 1$ **to** $N_{explore}$ **do**
         Add a new trajectory $\tau$ to $D_{explore}$ by sampling $\tau \sim \pi_{\mathcal{L}}$.
         Reshape all rewards in $\tau$ to be $\hat{r} = r + v_{\mathcal{F}}(\tau_{t+1}) - v_{\mathcal{F}}(\tau_t)$
      **end for**
      Train $\pi_{\mathcal{L}}$ using policy gradient with an additional cross entropy loss to $\pi_{\mathcal{F}}$ when $v_{\mathcal{F}}(\tau_t) \geq v$.
   **end for**

---

## 6 EXPERIMENTS

In order to evaluate ELF Distill, we compare it against the baselines in Section 5 on several challenging Minigrid (Chevalier-Boisvert et al., 2018) and Vizdoom (Wydmuch et al., 2019) environments with partial information. Figure 2 shows the results with diagrams of the environments.

The goal of the Minigrid environments is to reach a door that has the same color as a set of balls hidden in various locations in the environment. Observations are provided as a restricted top-down

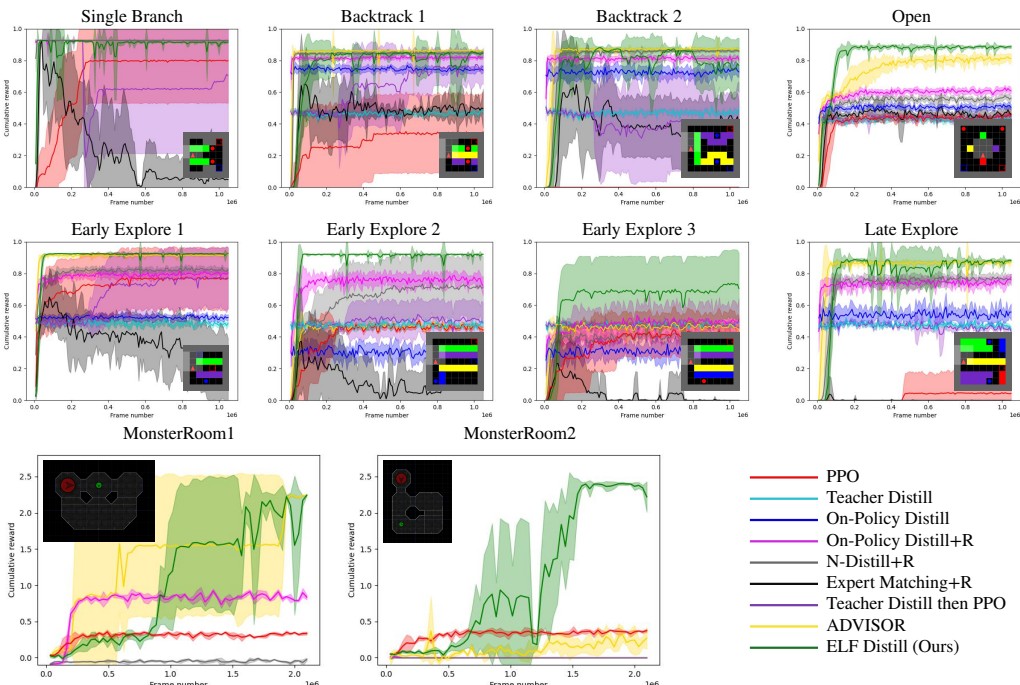

Figure 2: ELF Distill compared against seven baselines on eight Minigrid and two Vizdoom environments designed to require various levels of deviation from an impossibly good expert's advice. The inset image of each plot shows a diagram of the environment.

view of the gridworld. In this setting memory is handled for the agent by automatically remembering the color of the last ball seen and presenting as an additional observation variable. See Appendix B for details.

The goal of the Vizdoom environments is to navigate safely to an exit point without getting blown up by a cyborg demon. Each map has two exits, one of which is randomly guarded by the monster. The agent must find a window that allows it to see which exit the monster is guarding and take the alternate route. Unlike Minigrid, agents in these Vizdoom environments are provided with a first-person view from the player's perspective, and must remember relevant past observations using a recurrent network. See Appendix C for details.

From the experiments shown here, we can see ELF Distill performing comparably or better than all other baselines. Note that ADVISOR also performs quite well, except for a few environments (Early Explore 2, Early Explore 3, Monster Room 2) where the issues pointed out in Appendix E severely impact performance.

## 7 CONCLUSION

Imitation learning remains a powerful tool for sequential decision making problems. Unfortunately when experts have more information than the policies that learn from them, the experts may be impossibly good, and their behavior and performance cannot be replicated by any learning algorithm. We have shown that in these settings, it may still be possible to recover the optimal policy in the limited information space using only the expert's advice, but that this is not guaranteed as we have shown by providing counter-examples. To address this, we have introduced ELF Distillation, a training method that uses the estimated value function of a policy trained from expert demonstrations to guide exploration for a second reinforcement learning agent. We have shown that this algorithm outperforms several distillation baselines that incorporate both environmental reward and expert demonstrations on a set of challenging minigrid and vizdoom environments.

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

APPENDIX

## A  PROOF OF THEOREM 1

Recall that the policy $\widetilde{\pi}_\mathcal{L}$ is computed using the empirical distribution of a dataset $D$ sampled from a rollout policy $\pi_D$:

$$\widetilde{\pi}_\mathcal{L}(a|\tau_\mathcal{L}) = \frac{\sum_i^{|D|} \mathbb{1}_{\tau_{\mathcal{L}i}=\tau_\mathcal{L},a_{\mathcal{E}i}=a}}{\sum_i^{|D|} \mathbb{1}_{\tau_{\mathcal{L}i}=\tau_\mathcal{L}}} \tag{4}$$

We want to show:

*The empirical policy $\widetilde{\pi}_\mathcal{L}$ will recover $\pi_\mathcal{L}^*$ as $|D| \to \infty$ iff the following conditions hold:*

1. *(Coverage)* $\rho_{\pi_D}(\tau_\mathcal{L}^*) \neq 0 \ \forall \ \tau_\mathcal{L}^* \in P_{\pi_\mathcal{L}^*}$
2. *(Correctness)* $\mathbb{E}_{\tau_\mathcal{S} \sim \pi_D} \pi_\mathcal{E}(O_\mathcal{E}(\tau_\mathcal{S})) O_\mathcal{L}(\tau_\mathcal{L}^*|\tau_\mathcal{S}) = \pi_\mathcal{L}^*(\tau_\mathcal{L}^*) \ \forall \ \tau_\mathcal{L}^* \in P_{\pi_\mathcal{L}^*}$

*Proof.* We first show that the coverage condition is necessary, and then show that the correctness condition is necessary and sufficient when the coverage condition holds.

*The coverage condition is necessary:* If the coverage condition does not hold, for at least some observation histories $\tau_\mathcal{L}$, the denominator of Equation 4 will be:

$$\sum_i^{|D|} \mathbb{1}_{\tau_{\mathcal{L}i}=\tau_\mathcal{L}} = 0$$

This will result in an undefined distribution for $\widetilde{\pi}_\mathcal{L}$ at some trajectories that are encountered while acting according to $\pi_\mathcal{L}^*$. Undefined behavior is not optimal according $\pi_\mathcal{L}^*$ so this shows that the coverage condition is necessary to recover $\pi_\mathcal{L}^*$.

*The correctness condition is necessary and sufficient when the coverage condition holds:* Because the dataset $D$ is sampled according to $\pi_D$, as $|D| \to \infty$, the empirical distribution of expert recommendations in Equation 4 becomes:

$$\widetilde{\pi}_\mathcal{L}(\tau_\mathcal{L}) = \mathbb{E}_{\tau_\mathcal{S} \sim \pi_D} \pi_\mathcal{E}(O_\mathcal{E}(\tau_\mathcal{S})) O_\mathcal{L}(\tau_\mathcal{L}|\tau_\mathcal{S}) \ \forall \ \tau_\mathcal{L} \in P_{\pi_D}$$

If the coverage condition holds, then $P_{\pi_\mathcal{L}^*} \subseteq P_{\pi_D}$. This allows us to extend the expression above to $P_{\pi_\mathcal{L}^*}$, the set of trajectories with nonzero probability under $\pi_\mathcal{L}^*$

$$\widetilde{\pi}_\mathcal{L}(\tau_\mathcal{L}) = \mathbb{E}_{\tau_\mathcal{S} \sim \pi_D} \pi_\mathcal{E}(O_\mathcal{E}(\tau_\mathcal{S})) O_\mathcal{L}(\tau_\mathcal{L}^*|\tau_\mathcal{S}) \ \forall \ \tau_\mathcal{L}^* \in P_{\pi_\mathcal{L}^*}$$

This means that if the correctness condition holds:

$$\widetilde{\pi}_\mathcal{L}(\tau_\mathcal{L}) = \mathbb{E}_{\tau_\mathcal{S} \sim \pi_D} \pi_\mathcal{E}(O_\mathcal{E}(\tau_\mathcal{S})) O_\mathcal{L}(\tau_\mathcal{L}^*|\tau_\mathcal{S}) = \pi_\mathcal{L}^*(\tau_\mathcal{L}^*) \ \forall \ \tau_\mathcal{L}^* \in P_{\pi_\mathcal{L}^*}$$

∎

This shows that the agent-optimal policy can only be recovered if the distribution over labels generated by the expert $\pi_\mathcal{E}$ under the sampling distribution of the state space, match the agent-optimal policy $\pi_\mathcal{L}^*$ wherever the states map to $\tau_\mathcal{L}^*$.

## B  MINIGRID ENVIRONMENTS, MODEL AND TRAINING DETAILS

### B.1  ENVIRONMENT

As stated in Section 6, the goal of the minigrid environments is to reach the door that matches the color of one or more balls placed in the scene. At the beginning of each episode all balls in the scene set to be either blue or red, and they remain that color for the entire episode. There is one door per color and it's location is fixed for all time. The expert knows in advance the color of the balls and provides the agent with direct supervision to reach the correct door, but does not consider

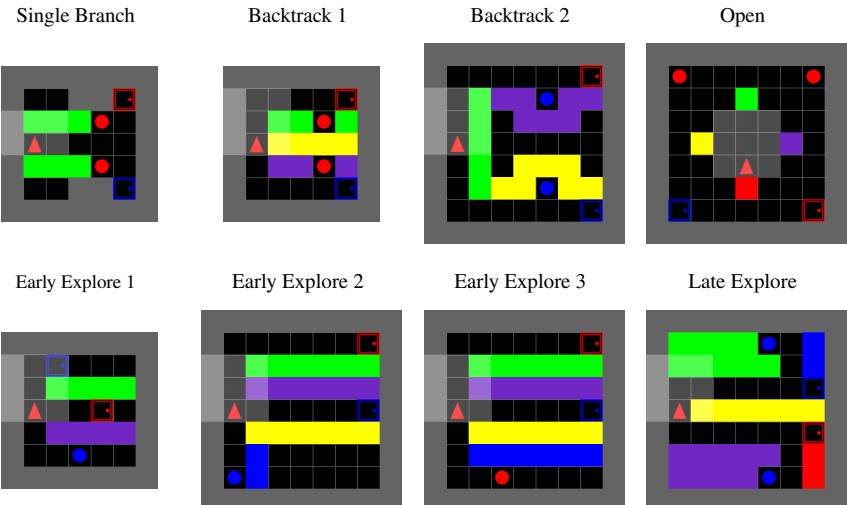

Figure 3: Minigrid Maps

the agent's ignorance of the ball color. Without taking further exploratory action to find the balls first, the agent cannot determine which door to go to and will always underperform compared to the expert. In order to succeed, the agent must deviate from the instructions provided by the expert in order to gather the information necessary to complete the task.

The agent gets a reward of $1 - 0.01|\tau|$ for entering the correct door for an episode. During training the expert demonstrations are provided by computing a shortest line path from the agent to the correct door without bothering to visit locations where the balls are located.

Visibility is strictly limited and the agent can only view the $3 \times 3$ tiles directly in front of them. While these environments are small, they are quite challenging due to the sparse reward and the need to automatically learn the color-matching behavior.

### B.2  MODEL

All minigrid methods train the same small model that takes a 3x3 grid with two channels representing object type (wall, door, etc) and a single color. The model automatically remembers the color of balls that it has seen in the past. The field of view of the agent is a 3x3 grid of tiles. The input to the network is provided as integer indices, so we use three embedding layers to construct a 3x3 16-channel representation of the observed object type, another 3x3 16-channel representation of the observed object color and a 1x1 16-channel representation of the remembered ball color (a third "grey" color is used when the ball has not been observed yet). The feature for the remembered ball color is tiled to 3x3 and these features are added together and flattened. The model then uses two fully connected layers with 256 channels each and ReLU activations. A policy head consists of another 256-channel linear layer followed by a Tanh activation and a linear projection to the number of actions 7. A value head is the same except it projects to a single value. ELF distill trains two policies, so it has one network for each. ADVISOR has an additional auxilliary head.

### B.3  TRAINING

Each method was trained on $2^{20} \approx 1M$ frames. When training ELF Distill, both the follower and explorer were trained on $2^{19} \approx 500K$ frames, so that the total frames observed during training was equal to the other methods. All methods were trained with 10 different random seeds in each environment. PPO (Schulman et al., 2017) was used as the loss function to maximize reward in all algorithms with a $\widehat{r}_i$ term.

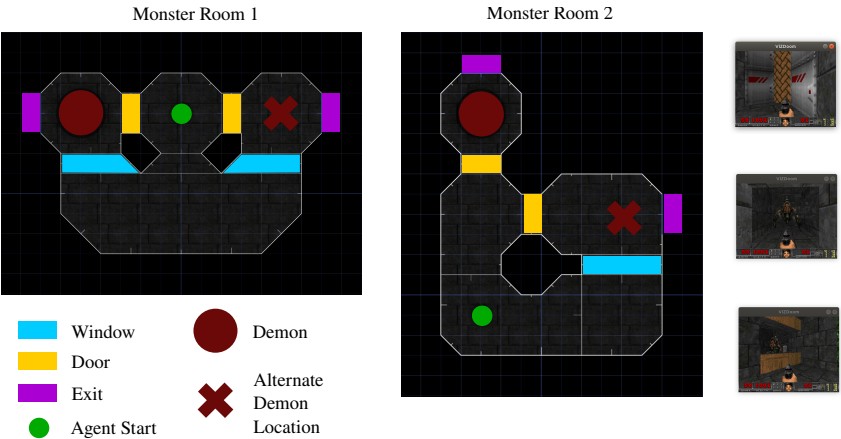

Figure 4: Vizdoom Maps

# C  VIZDOOM ENVIRONMENTS, MODEL AND TRAINING DETAILS

## C.1  ENVIRONMENTS

Our two VizDoom (Wydmuch et al., 2019) were built with the free SLADE Doom editor. The goal of these environments is to escape the room without getting destroyed by a cyborg demon. The agent must act using first-person visual data, and memory of the past. The native resolution of the environment is $320 \times 240$ RGB pixels which we rescale to $84x84$ grayscale channels. The game engine runs at a very high frequency, so we use a frame skip of 8 to avoid making decisions at too fine of a granularity. The agent has access to four actions: WALK FORWARD, TURN LEFT, TURN RIGHT and USE which both opens doors and pushes the switch to end the level. The damage settings have been tuned so that a single hit from the monster guarantees instant death. Although the agent can be seen carrying a pistol, it is purely decorative, and the agent has no ability to fight back against the monster.

The agent gets a reward of 2 for completing the level successfully, a reward of -2 for getting blown up, and a reward of 0 if neither happens before the maximum number of 72 frames. We also give the agent an exploration bonus based on how far it moves away from the nearest location it has been before, and a small negative reward of -0.001 at each time step. We also use early-termination, stopping an episode if the agent tries to walk forward, but makes no progress (hitting forward while pointing directly at a wall), switches between looking left and looking right three times in a row, or if the agent pushes the USE button twice in a row when not in front of a door or switch. These help avoid long trajectories of meaningless behavior in early episodes. If any of these early termination conditions are triggered, the episode ends with a small penalty of -0.05 for the agent.

## C.2  TRAINING

Each method was trained on $2^{21} \approx 2M$ frmes. As with Minigrid, we train each component of ELF Distillwith half that number so that the total number of training frames are comparable. All methods were trained with 3 different random seeds in each environment. As with Minigrid, PPO was used as the loss function to maximize reward in all algorithms with a $\hat{r}_i$ term.

# D  DISCUSSION OF RESULTS

## D.1  MINIGRID

Our eight MiniGrid Environments belong to five categories. In all environments we colored the walls so that the agents with their very limited field of view could find their way around.

**Single Branch** was designed to mimic Example I in Figure 1. In order to reach any door, the agent must pass by the balls. This is the easiest case, and the only one in which Teacher Distill can do better than 0.5. PPO is often successful here, but often takes longer to reach the goal than methods

that use the expert. Here we see a common failure mode of Expert Matching+R in that the agent learns a state-action loop for which the agent agrees with most of the expert's actions, but disagrees with a few of them. This allows the agent to continuously cycle and gain infinite reward according to it's reshaped reward objective that attempts to maximize agreement with the expert while failing to complete the task.

**Backtrack** are two environments, one larger than the other designed to mimic Example II in Figure 1. This time a ball is placed near each door so that if the agent heads to the wrong one initially, it can backtrack to the other. On-Policy Distill often reaches the correct goal, but does not perform as well as reward seeking approaches due to ambiguous expert advice before the balls have been observed. We also see Teacher Distill then PPO perform well in the smaller version, but it takes a long time to learn. PPO performs much worse on this task. ELF Distill, ADVISOR, On-Policy Distill+R and N-Distill+R all perform very well on these problems, with ELF Distill underperforming the others in a few cases.

**Early Explore** are three environments of different sizes designed to mimic Example III in Figure 1. Here the ball is placed on a separate hallway that is not on the path to any door. An algorithm must incorporate environmental rewards to be successful here. Here ELF Distill clearly dominates all other methods, although the problem becomes more difficult the futher the agent has to explore from states suggested by the expert. ADVISOR performs well on the first environment, but fails at the other two due to the issues pointed out in Appendix E.

**Late Explore** is similar to Backtrack in that the balls are near the doors. However in this case, they are far enough away that they will not be observed by an agent that walks directly to the door. Here ELF Distill and ADVISOR dominate, with On-Policy Distill+R and N-Distill+R also making progress.

**Open** places the balls and doors in opposite corners of the room with most of the space freely navigable. ELF Distill outperforms all others, while ADVISOR also makes progress.

## D.2 VIZDOOM

Our Vizdoom environments were designed to provide a more challenging visual scenario for training agents from impossibly good experts.

**Monster Room 1** Here the agent must exit a small room with two doors to enter a large room with windows where the agent can check for the existence of monsters. The agent can then exit the level by going back to the original room and opening the door to the empty room and proceeding to the exit. Only ELF Distill and ADVISOR are able to reliably complete this task.

**Monster Room 2** In the second Monster Room, the agent starts outside of the room with two doors. Because the expert knows where the monster is, it will always tell the agent to proceed to the room with two doors. This causes problems for ADVISOR, which will latch onto this predictable expert signal without knowing what to do once it has to make a decision about which door to enter. ELF Distill is the only method to reliably make progress on this task.

## E ADVISOR FAILURE CASE

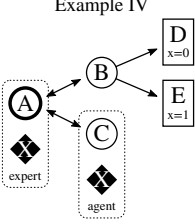

Figure 5: ADVISOR cannot recover the agent-optimal policy $\pi_{\mathcal{L}}^*$ in Example IV.

Example IV in Figure 5 provides a demonstration of a failure case for ADVISOR. In this example, nothing prevents the auxiliary policy $\pi_{aux}$ from replicating expert behavior at A, meaning the ADVISOR loss will strongly favor the imitation learning signal at this location which encourages the

transition from A to B. Unfortunately, however, the agent-optimal action is to first travel from A to C in order to learn the value of X, then backtrack to A and continue to the goal. Because ADVISOR is able to reproduce the expert's behavior at A, it will ignore the transition to C, and will be unable to gather the necessary information.

## F  LIMITATIONS

Because the explorer policy relies on value estimates from the follower policy, it is possible for biases in those estimates to cause poor explorer performance. This is especially true early in the training process before the follower's value function has had time to learn to differentiate good and bad states. This can be exacerbated by the fact that the follower's state coverage is determined by the explorer due to the switching policy used to collect training data for the follower. This can lead to situations where the follower underestimates the value of a particular state, which causes the explorer to visit it less often, which then causes the follower to get less training data for that state and therefore fail to recover from its initial error.

This suggests that it may be beneficial to use a training schedule that devotes more samples to the follower early on in order to provide better estimates of the value function, and more samples to the explorer later once this value function is well modelled.

Taken to an extreme, this would suggest training the follower alone for half of the training step budget, then freezing it and training the explorer. Unfortunately, this will not be ideal in many circumstances, as the explorer's policy is used to determine which states are necessary for the follower to learn. However, it may be possible to automatically decide when to allocate training resources to the follower or the explorer if the quality of the follower's value function can be estimated online.

## G  ACKNOWLEDGEMENTS

We would like to thank the members of the RAIVN and Robotics and State Estimation labs at the University of Washington for their support, discussion and feedback. We would also like to thank Luca Weihs for a valuable discussion of the ADVISOR algorithm.

