# OpenReview forum: "Impossibly Good Experts and How to Follow Them"
_ICLR.cc/2023/Conference — ICLR 2023 poster_

### Official Review · Reviewer_ZZSA · 2022-10-24

**Confidence:** 3
**Correctness:** 3
**Technical Novelty And Significance:** 3
**Empirical Novelty And Significance:** 2
**Recommendation:** 6

**Clarity, Quality, Novelty And Reproducibility:**

The proposed algorithm is well-motivated and original. The paper is well-written. My major concern is on the experimental side as all experiments are on conducted under one type of synthetic environment --- the minigrid. I would recommend this work to show the effectiveness of ELF Distillation in at least one real-world setting. Also, it would be nice to give some theoretical justifications/support for the proposed method (e.g., is the training of the follower guaranteed to assign high values to important states that the expert will never need to visit

**Strength And Weaknesses:**

Strength:
-

[+] The proposed training method ELF Distillation is novel and well-motivated.

[+] The work also empirically demonstrates that ELF Distillation performs better than a variety of strong baselines on a suite of minigrid environments.

Weakness:
-

[-] Regarding the training of the follower, I wonder how to guarantee that the follower will learn to give high values to critical states (e.g., visit the ball) which the expert will never need to visit. I did not see a specific term to motivate the follower to learn to assign high values to such states (despite that these critical states will appear in the training data)

[-] Regarding the experiments, current experiments are on minigrid with synthetic data, and I wonder how the proposed algorithm will perform in real-world scenarios.

Minor comment：
- Can you provide a concrete example of the problem-setting difference from Swamy et al. (2022). It is unclear from the description in the first paragraph of section 4
- What is the H function in Table 1？

**Summary Of The Paper:**

This work studied a interesting problem setting in imitation learning where experts have more information than the policies that learn from them. Thus, the experts may be impossibly good, and their behavior and performance cannot be replicated by any learning algorithm. This paper analyzed different existing approaches on this probelem setting and summarized one limitation: the existing approaches do not attempt to explicitly discover where following the expert yields high reward and where it does not. Based on this limitation, it proposed a training method, ELF Distillation, which uses the estimated value function of a policy agent trained from expert demonstrations to guide exploration for a second reinforcement learning agent. It further empirically demonstrates the power of the proposed method in minigrid environment. The experimental results show that the ELF Distillation outperforms several distillation baselines that incorporate both environmental reward and expert demonstrations.

The key insight in ELF Distillation is to train two policies jointly: a follower π_ϕ which attempts to learn how to follow the expert, and an explorer π_θ that attempts to maximize environmental reward using the follower’s value function as reward shaping. Training proceeds in an alternating fashion, first training the follower π_ϕ on new data generated by a combination of the explorer's action and the expert demonstration, then training the explorer π_θ on new data generated using the explorer.

Overall, this work proposed a novel and effective method in the problem setting where experts have more information available than the policy to be learned when making decisions. The paper is well-written, and the proposed method is also well-motivated. My major concern is on the experimental side as all experiments are on conducted under one type of synthetic environment --- the minigrid. I would recommend this work to show the effectiveness of ELF Distillation in at least one real-world setting.

**Summary Of The Review:**

Overall, this work proposed a novel and effective method in the problem setting where experts have more information available than the policy to be learned when making decisions. The paper is well-written, and the proposed method is also well-motivated. My major concern is on the experimental side as all experiments are on conducted under one type of synthetic environment --- the minigrid. I would recommend this work to show the effectiveness of ELF Distillation in at least one real-world setting.

---

> ### Author Response · Authors · 2022-11-15
> **Response to Official Review of ZZSA**
>
> Thank you so much for the detailed feedback, we very much appreciate your time and effort!  We have prepared an updated draft incorporating your suggestions and those of the other reviewers, with the exception of new experiments, which will be coming later this week.  We wanted to post this initial draft and the clarifying comments below to give you a chance to respond further if you wish.
>
> To address your specific comments and concerns:
>
> 1. “…My major concern is on the experimental side as all experiments are on conducted under one type of synthetic environment --- the minigrid. I would recommend this work to show the effectiveness of ELF Distillation in at least one real-world setting.” and “Regarding the experiments, current experiments are on minigrid with synthetic data, and I wonder how the proposed algorithm will perform in real-world scenarios.”  Thanks for this, we agree that the experiments are somewhat limited in the initial submission.  We are preparing follow-up experiments and will include details later this week.
>
> 2. “Regarding the training of the follower, I wonder how to guarantee that the follower will learn to give high values to critical states (e.g., visit the ball) which the expert will never need to visit. I did not see a specific term to motivate the follower to learn to assign high values to such states (despite that these critical states will appear in the training data)”  Sorry if this was not clear, when generating data for the follower, we first roll out according to the explorer for a random number of time steps, then switch to rolling out according to the follower (see the first inner loop in Algorithm 1 and the second paragraph under the bold “Elf Distillation” on page 7).  Starting the rollout using the explorer ensures that the trajectories will visit these critical states you mention.  Note however that we only train the follower’s value function for the time steps after the switch when the follower was in control so that it is not penalized for mistakes made by the explorer.  We have adjusted the explanation slightly to hopefully make this more clear.
>
> 3. “Can you provide a concrete example of the problem-setting difference from Swamy et al. (2022). It is unclear from the description in the first paragraph of section 4.”  Thank you for this note, we agree that this was unclear.  Swamy et al. is also concerned with settings where the expert has more information than the  learning agent.  However, Swamy is concerned with determining when expert performance can be achieved, where we are instead concerned with determining when we can recover the best agent policy possible in situations where reproducing the expert’s behavior may be impossible.  We have updated the first paragraph of section 4 to hopefully be more clear.
>
> 4. “What is the H function in Table 1?”  We apologize, H is cross entropy between the two distributions, we have included a mention of this in the latest draft.  Thanks for catching this!
>
> Thank you again for the feedback! We will follow up later this week with updated experiments. Feel free to respond with further questions or concerns in the meantime.

---

### Official Review · Reviewer_i2wK · 2022-10-24

**Confidence:** 3
**Correctness:** 4
**Technical Novelty And Significance:** 2
**Empirical Novelty And Significance:** 2
**Recommendation:** 6

**Clarity, Quality, Novelty And Reproducibility:**

As far as I can say the paper is clear, of high quality and the results are novel. The experiments should also be reproducible.

**Details Of Ethics Concerns:**

No concerns

**Strength And Weaknesses:**

Strengths:
- Interesting subject
- Overall good quality of writing


Weaknesses:
- The paper is not completely self-contained. I come from a different area and although I have worked on experts I found it difficult to follow some part mostly because of missing context.
- The example with the robot searching in an unknown environment reminds me of several settings from online algorithms. See for example "Exploring Unknown Environments" by Albers and Henzinger, SIAM J. Computing, 2000, or a sequence of papers on the "Cow path problem" and its generalisations. I personally would find it good if the current paper included a discussion on the differences/similarities between the models.

**Summary Of The Paper:**

The paper concerns itself with the situation in which some expert(s) may have access to more information than the learning agent. During training the expert has access to advice from the expert but this is not the case during testing.

After shortly describing the Behaviour Cloning and Dagger techniques, the paper investigates under what conditions these techniques confined the agent-optimal policy, and it is shown that in some settings the expert advice may not be sufficient for this purpose. Next it is investigated how to/whether to incorporate environmental reward into the learning algorithm -- to this end a new technique called ELF Distill is presented which is also shown to outperform previously known techniques (in most cases).

**Summary Of The Review:**

Overall I found the paper interesting. Given my background, I find it difficult to determine where exactly the results place in the landscape of current literature, but I found the studied paper, results and approach interesting and with the exception of some parts that I found hard to follow also generally well written. The results themselves and approaches employed are not very innovative, but in my point of view that is not necessarily required and the paper is still of interest.

---

> ### Author Response · Authors · 2022-11-14
> **Response to Official Review of i2wK**
>
> Thank you so much for taking the time to review our paper, we appreciate your feedback!  We have prepared an updated draft incorporating your suggestions and those of the other reviewers, with the exception of new experiments, which will be coming later this week.  We wanted to post this initial draft and the clarifying comments below to give you a chance to respond further if you wish.
>
> To address your specific comments and concerns:
>
> 1. “The paper is not completely self-contained. I come from a different area and although I have worked on experts I found it difficult to follow some part mostly because of missing context.”  Thank you for this feedback, it is valuable to hear.  We have updated the related work and section 5 with a few other methods as suggested by Aviv Tamar’s comment, that hopefully provide additional context about other related methods.  If there are specific items that need further contextualization, please let us know and we will address accordingly.
>
> 2. “The example with the robot searching in an unknown environment reminds me of several settings from online algorithms. See for example "Exploring Unknown Environments" by Albers and Henzinger, SIAM J. Computing, 2000, or a sequence of papers on the "Cow path problem" and its generalisations.”  Thank you for bringing this to our attention, the issue of exploration in unknown environments is very relevant to this problem.  In our setting we are forced to explore due our inability to perfectly follow the expert, and accomplish this using the built-in exploration of the underlying reinforcement learning component of the distillation algorithm (PPO in this case).  We have added the works you suggested, with another recent paper on exploration in deep reinforcement learning to the related work.  If there are more specific actions we can take to make this more clear, let us know and we will try to include them.
>
> 3. “Given my background, I find it difficult to determine where exactly the results place in the landscape of current literature,”  This is also useful feedback, thank you!  As mentioned above, we have added two other recent methods (ADVISOR and COSIL pointed out by Aviv Tamar above) to the related work and Section 5, hopefully these help to illustrate other recent methods and how we relate to them.  We are also considering further discussion of these related methods in the introduction, if this would be helpful, or if there is more we can do here, let us know!
>
> 4. “The results themselves and approaches employed are not very innovative…” Thanks for this, we received similar feedback from Reviewer tPto.  We feel that the primary novelty of our method is abandoning an explicit expert following term in favor of seeking states from which following the expert leads to strong performance.  While the components we use to accomplish this are not new, we hope that our novel combination of them here is both interesting and of significant practical relevance to people working in this area.  Hopefully the updated passages on ELF-Distill in Section 5 make this more clear.
>
> Thank you again for the feedback!  We will follow up later this week with updated experiments. Feel free to respond with further questions or concerns in the meantime.

---

> > ### Comment · Reviewer_i2wK · 2022-11-25
> > **Thanks**
> >
> > Thanks for the response to my comments.

---

### Official Review · Reviewer_tPto · 2022-10-28

**Confidence:** 3
**Correctness:** 4
**Technical Novelty And Significance:** 2
**Empirical Novelty And Significance:** 2
**Recommendation:** 6

**Clarity, Quality, Novelty And Reproducibility:**

* Clarity: The clarity of the overall paper was fine, but I thought the notation choice was a bit difficult to read (why use $**$ in the superscript?) and I also thought the language used to describe Example 2 on page 5 (and Example 3, but less so) could be clearer - in particular, for Example 2 - "the expert will travel from A to B when X = 0 and from A to C when X = 1" - based on Fig. 1's version of Example 2, this doesn't seem to match up.

* Quality: The quality of the paper was fine.

* Novelty: The algorithm seems to be novel though I am not extremely familiar with the related literature cited in the paper; the related work section seems pretty good. However as mentioned before, the ideas in the algorithm do not appear to be very novel in themselves.

* Reproducibility: It seems fairly easy to reproduce the paper though I have not done so myself.

**Strength And Weaknesses:**

* Strengths: The problem setting proposed is an interesting one, and a novel algorithm is provided which appears to perform favorably in some gridworld settings. The authors also provide some concrete examples of when various other natural algorithms to try may not work. They also provide a characterization of some conditions on environments for which an empirical distribution of expert demonstrations will converge to the optimal policy. Finally, the authors test their new algorithm on several conceptually sound environments that probe at natural difficulties involved with the problem setting.

* Weaknesses: The primary contribution of this paper, in my opinion, is the novel algorithm that they propose and test -- the counter-examples are fairly straightforward. However, the ideas in the algorithm do not appear to be that novel by themselves. This in itself would be fine if it were shown that the resulting algorithm works significantly better than other approaches on many kinds empirically, or if there were some interesting theoretical insight to pair with the improved algorithm, but the experiments are fairly limited and do not necessarily show a significant improvement over existing methods, though it seems clear that a trend exists that the mean performance of ELF is better on all these tasks. Also, Theorem 1 is not that interesting in my opinion (the conditions are fairly clear to see). Thus I am more lukewarm on accepting the paper.

**Summary Of The Paper:**

In this paper, the problem of imitating an expert with access to more information about the state space in a POMDP than a learner is considered -- these are called impossibly good experts. Existing methods for imitation learning like Behavior Cloning and the DAGGER algorithm are shown to fail with concrete counterexamples, and necessary and sufficient conditions are given for learning from an impossibly good expert. The paper proposes an algorithm to solve this problem called ELF Distillation, which makes use of existing techniques for making use of information from sub-optimal experts (the paper considers sub-optimal experts because environments can be constructed where the impossibly good expert is unhelpful) by combining expert demonstrations with reward signal from the environment itself. The main twist that the ELF algorithm proposes on top of these existing methods is to avoid balancing reward seeking behavior and expert following $\textit{uniformly}$ across all sequences of behavior; ELF instead proposes to jointly train a follower that follows the expert and an explorer which uses the follower's learned value function as a reward-shaping mechanism. This algorithm is tested empirically on grid-world based POMDP tasks and against an expert given appropriate information commensurate with the problem setting, for which it performs favorably compared to existing methods.

**Summary Of The Review:**

Overall, the paper is a decent contribution to the literature on following an impossibly good expert. My main concern is on the novelty/ sufficient amount of additional learnings present in the paper.

===== Post-Rebuttal ======

After seeing the update, I'm inclined to think that the additional experiments and framing of the contribution make the paper sufficiently interesting for publication. Thus I updated my score.

---

> ### Author Response · Authors · 2022-11-14
> **Response to Official Review of tPto**
>
> Thank you for your thoughts and insight, and for taking the time to thoroughly review our paper, your comments have been very valuable.  We have prepared an updated draft incorporating your suggestions and those of the other reviewers, with the exception of new experiments, which will be coming later this week.  We wanted to post this initial draft and the clarifying comments below to give you a chance to respond further if you wish.
>
> To address your specific comments and concerns:
> 1. “…the counter-examples are fairly straightforward.”  This is true, our goal here was to come up with the simplest examples we could think of to clearly spell out the pitfalls that can arise when using existing imitation learning algorithms.  While this is not a deep technical contribution, we feel it is important guidance for practitioners, who may use these methods without realizing the hidden dangers.
>
> 2. “However, the ideas in the algorithm do not appear to be that novel by themselves.”  While it is true that our method uses components of existing algorithms and ideas, ELF-Distill is a new technique in this domain that to our knowledge has not been presented elsewhere.  While the use of these existing components may make ELF-Distill seem less novel at first glance, we feel that the key idea of removing an explicit expert following term and instead seeking states where following the expert leads to good long-term performance is both novel and of significant practical utility to practitioners.  We have attempted to improve the presentation of these features in the new draft to highlight this contribution, especially in comparison to ADVISOR, a related paper which was added at the request of Aviv Tamar (see section 5 in the updated draft).
>
> 3. “…the experiments are fairly limited and do not necessarily show a significant improvement over existing methods, though it seems clear that a trend exists that the mean performance of ELF is better on all these tasks.”  We agree that the experiments in the initial submission are somewhat limited.  We are working on expanding these and will reply with an updated draft later this week.  Note however that in many of the existing Minigrid environments, ELF-Distill reaches greater levels of performance than all other methods.
>
> 4. “Also, Theorem 1 is not that interesting in my opinion (the conditions are fairly clear to see).”  Thank you for this feedback, if the other reviewers respond similarly, we may remove or deemphasize this.
>
> 5. “The clarity of the overall paper was fine, but I thought the notation choice was a bit difficult to read (why use ** in the superscript?)”  Thank you for this, we have received similar feedback from others and have updated the notation.  We now use the subscript L to refer to variables related to the agent, and E to refer to variables related to the expert.  If there are any additional points of confusion please let us know!
>
> 6. “…and I also thought the language used to describe Example 2 on page 5 (and Example 3, but less so) could be clearer - in particular, for Example 2 - "the expert will travel from A to B when X = 0 and from A to C when X = 1" - based on Fig. 1's version of Example 2, this doesn't seem to match up.”  Thank you for this, we believe the confusion arose because Figure 1 was mislabeled.  The labels previously said Example II, Example III and Example IV, but should have said Example I, Example II and Example III.  Apologies for this, we have corrected this in the new draft.  If this does not clear up the confusion, let us know and we will try to address this further.
>
> Thank you again for the feedback!  We will follow up later this week with updated experiments.  Feel free to respond with further questions or concerns in the meantime.

---

> > ### Comment · Reviewer_tPto · 2022-11-17
> > **Response to the Rebuttal**
> >
> > Thanks for the update!
> >
> > I will look forward to the updated experiments (which I assume will contain experiments comparing against ADVISOR?).
> > The section regarding Examples I-III reads much smoother, thanks. I also appreciated the added section on ADVISOR, which indeed does clarify an interesting (seemingly previously missing?) insight regarding the ELF algorithm. One remark about Figure 4 in Appendix D: It would be good to clarify the reason why the optimal action is to transition to C (e.g., to get the value x) -- this could be easily fixed by saying something like "transition to C *first*,", the way it was originally written, it seemed like you were saying C was a terminal state (which is obviously not the case if you have Examples I-III in mind, but is harder to read without that context, as might be the case for someone skipping to the appendix). Also regarding the comparison to ADVISOR and Fig. 4 in Appendix D, it would be helpful to provide an example of a case where it is hard for the auxiliary policy to imitate the expert (so that the weighting is small), I was a bit confused about when the weighting term will be small and how small it will be.
> >
> > Regarding Thm. 1, I think it's fine to keep in, though I wonder if something more concrete could be said about the advantage of expert value estimation and avoiding expert following (weighted between imitation and RL loss as in ADVISOR) in terms of efficiency of learning beyond Example IV being sub-optimal (and it would be good to make the sub-optimality in Example IV more concrete). It would be nice to maybe construct a family of examples parameterized by size (using a modified version of Example IV) to explicitly/concretely show some worst-case efficiency gap for the behavior of any algorithm that tries to blend RL+imitation losses compared to the expert value learning approach.

---

### Public Comment · ~Aviv_Tamar1 · 2022-11-10
**Missing relevant related works**

This is an interesting problem, which several recent works already tried to tackle. Please see:
1. Nguyen, Hai Huu, et al. "Leveraging Fully Observable Policies for Learning under Partial Observability." 6th Annual Conference on Robot Learning.
2. Warrington, Andrew, et al. "Robust asymmetric learning in pomdps." International Conference on Machine Learning. PMLR, 2021.
3. Weihs, Luca, et al. "Bridging the imitation gap by adaptive insubordination." Advances in Neural Information Processing Systems 34 (2021): 19134-19146.

It would be good to situate ELF in this space.

---

> ### Author Response · Authors · 2022-11-14
> **Thanks!**
>
> Thank you for adding this!  We missed these when putting together the initial submission, but a collaborator also pointed these out to us recently.  Fortunately, despite this, ELF is distinct from these methods in important ways.  ELF does not require a differentiable expert which is required by Warrington et al.  Also, rather than incorporating a uniform imitation learning signal (Nguyen et al.) or attempting to blend between imitation learning and RL based on the agent’s ability to immediately follow the expert (Weihs et al.), ELF encourages following the expert when doing so leads to long-term success.  We have added these methods to the related work and section 5 and will attempt to incorporate Nguyen et al. and Weihs et al. into the experimental baselines when we release new experiments later this week.

---

### Author Response · Authors · 2022-11-14
**First rebuttal update posted.**

Thanks to all reviewers for their thoughtful comments and analysis.  We have uploaded a new rebuttal version which incorporates various concerns addressed by the reviewers.  The one thing that is NOT included yet is new experiments which will be coming later this week.  See individual responses below for how these changes are meant to address your comments.  Feel free to either respond now, or wait until later in the week for new experiments.  Thanks again!

---

### Author Response · Authors · 2022-11-19
**Thanks!**

We are uploading our latest rebuttal version now with ADVISOR results on Minigrid, and two new VizDoom environments with challenging first-person visual inputs.  Thank you to all of the reviewers for your thoughtful comments!  Your contributions made our paper better.

---

### Decision · Program_Chairs · 2023-01-20

**Decision:**

Accept: poster

**Justification For Why Not Higher Score:**

I wouldnt mind if this was bumped up to a spotlight. The main reason for a "poster" recommendation was the reviewer scores (all of which are "6")

**Justification For Why Not Lower Score:**

The paper's setting and algorithmic novelty warrant inclusion.

**Metareview: Summary, Strengths And Weaknesses:**

The paper proposes a (relatively) new setting: POMDPs which also have an expert which has more information about the hidden Markov state than the learner. The algorithm proposed is novel as well, and differs from the 2-3 other works (that were pointed out by a third party but not the reviewers).

Overall the reviewers agree the paper is well written and the novelty of the algorithm and the interesting-ness of the setting make this a worthy candidate to be included in ICLR as a poster.

**Note From Pc:**

if the above contains the word "oral" or "spotlight" please see: "oral" presentation means -> notable-top-5% and "spotlight" means -> notable-top-25%. As stated in our emails, we are disassociating presentation type from AC recommendations

**Summary Of Ac-Reviewer Meeting:**

N/A